# Application of Wastewater Reuse with Photocatalyst Prepared by Sol-Gel Method and Its Kinetics on the Decomposition of Low Molecular Weight Pollutants

**DOI:** 10.3390/ijerph17124203

**Published:** 2020-06-12

**Authors:** Jinwook Chung, Seungjoon Chung, Gyuyoung Lee, Yong-Woo Lee

**Affiliations:** 1R&D Center, Samsung Engineering Co. Ltd., 41 Maeyoung-Ro, 269 Beon-Gil, Youngtong-Gu, Suwon, Gyeonggi-Do 16523, Korea; chung.j@samsung.com (J.C.); phd.chung@samsung.com (S.C.); young8.lee@samsung.com (G.L.); 2Department of Chemical and Molecular Engineering, Hanyang University, 55 Hanyangdaehakro, Sangrok-Gu, Ansan, Gyeonggi-Do 15588, Korea

**Keywords:** electronic wastewater, low-molecular-weight (LMW) compounds, photocatalytic media, sol-gel method

## Abstract

The development of immobilized photocatalyst as a strategy for problematic electronics wastewater reuse is described in this study. The strategy was to perform separate rinsing, mostly consisting of low molecular weight compounds, and to decompose them with a simple process, based on the advanced oxidation process (AOP). Extensive studies were performed on the preparation conditions of immobilized photocatalysts by sol-gel method under various amount of precursor and support, water to precursor ratio, pH, aging time, and calcination conditions. The optimized preparation conditions were chosen by measuring removal efficiencies of isopropyl alcohol as a representative target compound with supportive SEM and XRD analyses. Removal efficiencies with photocatalyst and UV irradiation in synthetic wastewater simulating electronics wastewater were evaluated over time. Removal efficiencies of alcohol, acetone, ethanol, and acetaldehyde reached 97.2%, 71.2%, 99.0%, and 99.0%, respectively, in 2 h. Reaction constants of each compound were determined by fitting experimental data to the first order kinetic equation and the trial and error method with consecutive reaction pathway. As analysis results of reaction constants, UV with prepared photocatalyst was found to be effective and the decomposition of acetone was found to be the rate-determining step. The immobilized photocatalyst developed in this study would be useful for application of wastewater reuse with high removal efficiencies, mild preparation conditions, and mechanical stability.

## 1. Introduction

According to a Korean government report, the most water-intensive sectors are textile, pulp, and electronics industries, and the water consumption in those industries has gradually increased over the years [1]. In particular, the electronics industry has undergone a severe water shortage problem due to its recent rapid growth and limited water supply. The production facilities in electronics industries, such as semiconductors and display manufacturing, use various toxic chemicals, mostly low molecular organic compounds, such as isopropyl alcohol and acetone [2]. These compounds are not completely removed by conventional wastewater treatment processes, such as reverse osmosis and biological treatment. To meet stringent water quality needs in the electronics industry, additional treatments, such as advanced oxidation processes (AOPs), have been considered [3]. AOPs utilize hydroxyl radicals with a greater oxidation potential than chemical oxidants for decomposing low molecular weight compounds. Service water, so-called ultrapure water, in the electronics industry requires high purity to avoid contamination of the products. To meet the required purity, tap water that is readily purified by several processes, is further treated with a complex combination of unit processes, such as reverse osmosis, ion exchange, and ultrafiltration. This combination of processes demands a large footprint and capital expense depending on the raw water quality and required purity. Thus, the following strategy for managing wastewater and its reuse stream has been considered. Once a waste stream consisting of low molecular weight organic compounds, e.g., rinsing water, is appropriately separated, it can be treated by a simple AOP-based process for reuse [4]. The development of efficient AOP technology has a great impact on the reuse of wastewater from the electronics industry.

There are many different methods used in AOPs, such as UV, UV + O_3_, UV + H_2_O_2_, and UV + photocatalysts [5]. UV cannot completely degrade organic compounds into small molecules, such as water and carbon dioxide. UV + O_3_ and UV + H_2_O_2_ have disadvantages in the use of additional chemicals. Many studies have focused on the use of UV/photocatalysts due to strong photochemical activity and semi-permanent reusability without chemical addition. Photocatalysts are semiconducting metal oxides, such as Al_2_O_3_, WO_3_, TiO_2_, and ZnO, which are able to produce hydroxyl radicals assisted by photons [6]. The most common photocatalyst is TiO_2_ due to its sufficient hydroxyl radical production capability and photo-chemical stability. There have been many studies on the immobilization of photocatalysts because it is not easy to recover small suspended photocatalyst particles for reuse [7]. Photocatalysts have been immobilized on supports, such as glass, sand, Teflon, activated carbon, and ceramics, by the sol-gel method, chemical vapor deposition, and physical vapor deposition [8]. Among these, the sol–gel method is a promising technique for immobilization of photocatalysts due to its mild conditions, simple process, and low cost [9].

The study on the removal of low molecular weight compounds by photocatalyst for reuse of electronics wastewater was relatively unexplored, while the development of an efficient AOP technology has a great impact on the expansion of the reuse process. In addition, it is challenging to immobilize photocatalysts in the sol-gel method with good morphology, photochemical activity, and mechanical stability due to many factors, such as concentration of the precursor, water to precursor ratio, pH, aging time, and calcination conditions [10,11,12].

Authors suggested the immobilization of photocatalysts on hollow beads by chemical vapor deposition in the previous study for fluidized bed operation [13]. In this study, an extensive and systematic study was performed to optimize various factors in preparation conditions via efficient sol-gel routes to immobilize a photocatalyst on a hollow bead, as described in Scheme 1. In addition, the removal efficiencies of UV + prepared photocatalyst were compared with those of UV only. Furthermore, the reaction kinetics to describe experimental data were investigated and discussed.

## 2. Materials and Methods

### 2.1. Preparation and Characterization of Photocatalytic Media

Ceramic hollow beads (DFL, Minmet Refractories Co, Cenosphere, China) were used as a support to bind photocatalysts in the fluidized bed reactor. These beads were mainly composed of alumina (Al_2_O_3_) and silica (SiO_2_) and were 40–300 μm in diameter with particle density of 0.9–1.0 g/mL. Photocatalyst was immobilized in three steps. In the first step, the reactant and support were homogenized. Forty ml of methanol and nitric acid were mixed to adjust the pH of 1–7. Various amounts of hollow beads (10–35 g) and titanium butoxide (4–16 mL) were mixed and stirred for 3 min. The next step was the deposition of a TiO_2_ layer on the support by hydrolysis and condensation reaction. Various amounts of distilled water (0.6–3.6 mL) were injected into the mixture with stirring for different aging times ranging from 0.5 to 2 h. After completion of the reaction, unreacted reactants were decanted off and beads were retained by stainless steel mesh. After drying at the ambient temperature, beads were dried in the oven at 80 °C for 3 h. The final step was building the crystalline structure of TiO_2_ on the surface of beads. The dried beads were calcined at different temperatures ranging from 400 to 800 °C for a calcination time of 1 to 4 h. The preparation conditions of the photocatalytic media are listed in Table 1. The prepared photocatalytic media in different preparation conditions was analyzed using field emission scanning electron microscopy (FE-SEM, S-4800, Hitachi, Co., Japan) and an X-ray diffractometer (XRD, D/Max 2500 PC, Rigaku Co. Japan). FE-SEM specimens were attached to the sample holder using double sided carbon tape and subsequently coated by platinum before analysis. The X-ray emission angles ranged from 20 to 55 degrees with a Cu-kα source.

### 2.2. Evaluation of Removal Efficiencies of Target Compounds

As described in Figure 1, a modified drying oven (900 × 900 × 900 mm^3^ in size) was used as an experimental apparatus for photocatalytic activity of prepared media. A quartz dish that was 160 mm in diameter and 50 mm in height was used as a reaction vessel to hold 1 L of synthetic wastewater and 1.125 g of photocatalytic media. A UV lamp (Atlantic Ultraviolet, GPH357T5L-17W, USA) was installed on top of the oven for irradiation of the photocatalyst. The distance between the UV lamp and reaction vessel was 50 mm. Synthetic wastewater was prepared to simulate wastewater for reuse in the electronics industry by using 1000 μg/L of spiking isopropyl alcohol, acetone, ethanol, and acetaldehyde. In addition, a removal experiment with UV irradiation was performed for comparison. The concentrations of target compounds were adjusted to 1000 μg/L. Samples from the reaction vessel were taken with time and the concentrations of target compounds were quantified using gas chromatography with mass spectroscopy (GC/MS, gas chromatography 7820A, mass spectrometry 5975, Agilent) coupled with purge and trap (Stratum, Teledyne, Tekmar, USA). The operational conditions of purge and trap were as follows. Sample and purge temperatures were 40 °C. Nitrogen gas was used for purging with a flow rate of 100 mL/min for 8 min. The following conditions were used for GC-MS analysis. The target compounds in the sample were separated with a DB-624 column (Agilent). The temperatures of the inlet and detector were 100 °C and 250 °C, respectively. The oven temperature was held at 35 °C for 7 min and 46 °C for 5 min with a ramping rate of 3 °C/min. Standard solutions for target compounds were prepared by series dilution of stock solution prepared by American Chemical Society (ACS) grade chemicals (Sigma Aldrich). Deviation from calibration curve was checked in every 3–5 samples and kept less than 10%. The concentrations of target compounds in the blank eluent from hollow beads, dissolved in water from air, and in distilled water were measured to correct the baseline as described in the literature [13].

## 3. Results and Discussion

### 3.1. Preparation and Characterization of Photocatalysts

Immobilized photocatalysts under various conditions were prepared and removal efficiencies of a single constituent (isopropyl alcohol, as a representative for target compounds) were evaluated for selection of preparation conditions. These preliminary tests were conducted in high loading conditions (10 mg/L of isopropyl alcohol) to amplify sensitivity of removal in different preparation conditions with a single constituent rather than testing with multiple constituents (1 mg/L of each). The removals of isopropyl alcohol in various preparation conditions are presented in Figure 2a–f. Supportive SEM images and XRD patterns are presented in Figure 3 and Figure 4, respectively.

Figure 2a indicates that a decrease in the amount of support with a fixed amount of precursor led to higher removal possibly due to the increase in active sites of the support for anchoring more precursors. In Figure 2b, an increase in the amount of precursor with a fixed amount of support tends to raise the amount of removal due to the formation of a thicker coating layer (data not shown). As shown in Figure 2c, the removal increased with aging time and reached a maximum at 60 min. A defective layer of photocatalyst is shown in Figure 3b (aging time = 120 min), while a defect-less layer is shown in Figure 3a (aging time = 60 min). The detachment of the excessive coating layer was observed after calcination of photocatalyst with more than 60 min of aging time. Thus, the 60 min was chosen as an optimal aging time with maximal thickness of 100–150 nm (measured in SEM analysis). The amount of precursor, support, and aging time were variables affecting the mechanical stability as well as photochemical ability of photocatalyst.

In addition, the pH and amount of water in the sol-gel reaction plays an important role in controlling the reaction rate of hydrolysis and condensation [14]. Figure 2d,e indicate that a lower pH and more water addition favor a higher removal. The SEM results in Figure 3d,f show that a high pH and more water addition produced agglomeration on the surface of the photocatalyst layer. The water to precursor molar ratio (r = [H_2_O] / [Ti(OBu)_4_]) in the sol-gel process strongly affected the morphology of the photocatalyst. It was reported that the Ti(OBu)_4_ precursor was fully hydrolyzed to Ti(OOH)_4_ in a high ratio (r = 4), while it was partially hydrolyzed to Ti(OH)_r_(OBu)_4-r_ in a low r ratio (r < 4). The fully hydrolyzed precursor would lead to a more crosslinked structure [15]. In addition, it has been reported that a high r ratio favored a smaller size of particles, which is in agreement with the results of this study [11]. Amount of water was determined to have r close to 4 (amount of water = 2.4 mL) with less agglomeration.

In the literature, the anatase phase was easily developed at a pH above 4.5, while the rutile phase was at a lower pH between 2.5–4.5 [16]. However, it was noted that the anatase phase could be formed at a low pH and low calcination temperature in another report [17]. In addition, it was claimed that a low pH (about 2) led to a small spherical nano-sized particle, while a higher pH led to bigger particles [11]. The plausible explanation for this result was that the positively charged surface of the support and precursor in a low pH exhibited a repulsive force reducing the reaction rate of the precursor. This relatively slow reaction rate in a lower pH would result in a uniform surface with smaller grain than the discrete surface in a higher pH. This was in agreement with the formation of agglomerations at a higher pH in this study and in the literature [12]. The preparation conditions of the sol-gel process (amount of support = 15 g, amount of precursor = 12 mL, amount of water = 2.4 mL, and aging time = 60 min) were chosen accordingly based on the experimental investigations.

Calcination conditions significantly affected the crystalline structure of the photocatalyst. The removal tended to decrease with an increase in sintering temperature from 500 to 800 °C, as shown in Figure 2f. Calcination time does not significantly change after 2 h, as shown in Figure 2g. The XRD patterns shown in Figure 4a indicate that the intensity peak (25°) of the anatase phase appeared at sintering temperatures of 500–700 °C, while those of the rutile phase peak (37.5°) appeared at 800 °C. In Figure 4b, the anatase phase peak appeared after 2 h of calcination. The higher removal in low calcination temperatures (450–500 °C) was due to the formation of the anatase phase with a higher purity. This calcination temperature was close to that in the literature (350–450 °C) where the anatase phase could be fully developed from the amorphous phase [10]. The calcination conditions were determined to have a high removal efficiency (calcination temperature = 500 °C and calcination time = 2 h). SEM images of immobilized photocatalyst are shown in Figure 3g,h with different magnifications. A uniform and smooth surface of the photocatalyst was successfully developed on the surface of media. The diameter of the prepared media was about 100 μm and the thickness of deposition layer was about 100 nm.

### 3.2. UV/Photocatalytic Removal and Reaction Kinetics

Removal efficiencies of target compounds with immobilized photocatalyst under UV irradiation are presented in Figure 5a–d. In addition, experimental removal efficiencies without photocatalysts (UV only) were included in the figures for comparison. The removal efficiencies of isopropyl alcohol, ethanol, and acetaldehyde with immobilized photocatalyst are clearly higher than those without photocatalysts at all times. However, the removal efficiencies of acetone by photocatalytic media were lower than those with UV only from 15 to 60 min, while they were higher at 120 min. To account for this discrepancy, reaction kinetics were calculated and compared. After trials of fitting experimental data to the rate equations from zero to second order kinetics, the first-order kinetics were determined as follows:(1)$$\frac{dC_{i}}{dt}=-k_{i,ap}C_{i}$$
(2)$$\frac{C_{i}}{C_{i0}}=e^{-k_{i,ap}t}$$
where *i* = *a*, *b*, *c*, *d* (*a*: isopropyl alcohol, *b*: acetone, *c*: ethanol, and *d*: acetaldehyde); and *k_i_* is the apparent reaction constant of the *i* compound.

The apparent reaction constants of target compounds are calculated and are listed in Table 2. The reaction constants of isopropyl alcohol and ethanol were determined with a high coefficient of determination (R^2^ = 0.99), while those of acetone and acetaldehyde were determined with a poor to moderate coefficient of determination (R^2^ = 0.91 − 0.97). The possible reasons for this poor correlation could have originated from the formation of byproducts. Thus, the consecutive reaction relationships of isopropyl alcohol, acetone, ethanol, and acetaldehyde were considered as shown in Figure 6 [18,19,20,21].

The reactions between target compounds and hydroxyl radicals involved in the reaction pathway would be described as follows:IPA(C_3_H_8_O) + 2·OH → Acetone(C_3_H_6_O) + 2H_2_O(3)
Ethanol(C_2_H_6_O) + 2·OH → Acetaldehyde(C_2_H_4_O) + 2H_2_O(4)
2Acetone(C_3_H_6_O) + 2·OH → 3Acetaldehyde(C_2_H_4_O) + H_2_O(5)

According to this reaction pathway, isopropyl alcohol decomposes to acetone, which decomposes to acetaldehyde. In addition, ethanol decomposes to acetaldehyde. Acetone and aldehyde are not only decomposed, but they are also produced during UV photolysis with or without a photocatalyst. When the first-order kinetics and the suggested reaction pathway are assumed, the rate equations are as follows:(6)$$\frac{dC_{a}}{dt}=-k_{a,ap}C_{a}$$
(7)$$\frac{C_{a}}{C_{a0}}=e^{-k_{a,ap}t}$$
(8)$$\frac{dC_{b}}{dt}=k_{a,ap}C_{a}-k_{b,co}C_{b}$$
(9)$$\frac{C_{b}}{C_{b0}}=e^{-k_{b,co}t}+\frac{k_{a}}{k_{b}-k_{a}}\left(e^{-k_{a,ap}t}-e^{-k_{b,co}t}\right)$$
(10)$$\frac{dC_{c}}{dt}=-k_{c,ap}C_{c}$$
(11)$$\frac{C_{c}}{C_{c0}}=e^{-k_{c,ap}t}$$
(12)$$\frac{dC_{d}}{dt}=k_{b,co}C_{b}+k_{c,ap}C_{c}-k_{b,co}C_{d}$$
(13)$$\frac{C_{d}}{C_{d0}}=e^{-k_{d,co}t}+\frac{k_{b,co}}{\left(k_{d,co}-k_{b,co}\right)}\left(e^{-k_{b,co}t}-e^{-k_{d,co}t}\right)+\frac{k_{c,ap}}{\left(k_{d,co}-k_{c,ap}\right)}\left(e^{-k_{c,ap}t}-e^{-k_{d,co}t}\right)\;+\frac{k_{a,ap}k_{b,co}}{\left(k_{b,co}-k_{a,ap}\right)}\left(\frac{e^{-k_{a,ap}t}-e^{-k_{d,co}t}}{k_{d.co}-k_{a,ap}}-\frac{e^{-k_{b,co}t}-e^{-k_{d,co}t}}{k_{d,co}-k_{b,co}}\right)$$
Here, the subscript a is isopropyl alcohol, b is acetone, c is ethanol, and d is acetaldehyde;

*k_i,ap_* is the apparent reaction constant; and *k_i,co_* is the reaction constant with the consecutive reaction relationship.

*k_b,co_* and *k_b,co_* were determined to have high coefficients of determination (R^2^) using the trial and error method with equations for *C_b_* and *C_d_* t. This coefficient was calculated using the following equations:(14)$${\mathrm{SS}}_{res}=\underset{i}{\sum }{\left(y_{i}-f_{i}\right)}^{2}$$
(15)$${\mathrm{SS}}_{tot}=\underset{i}{\sum }{\left(y_{i}-\bar{y}\right)}^{2}$$
(16)$$R^{2}=\frac{{\mathrm{SS}}_{res}\;}{SS_{tot}}$$
where, *y_i_* are experimental data and *f_i_* are calculated values.

The newly calculated reaction constants are listed in Table 2, which are used to describe experimental data under UV irradiation with and without photocatalysts, as shown in Figure 7. When the newly calculated reaction constants are used, the experimental data is described better with a higher coefficient of determination, as shown in Figure 7. This type of approach would be useful when apparent reaction constants are determined with a poor coefficient of determination. UV with prepared photocatalyst was found to be effective, because its reaction constants were higher than those without photocatalysts with a fairly good coefficient of determination. In addition, it was revealed that the UV irradiation with and without prepared media mostly followed the suggested reaction pathway with a consecutive relationship. The low removal efficiency of acetone in 15–60 min was caused by a consecutive reaction relationship, and the difference in reaction constants between isopropyl alcohol and acetone. If decomposition of acetone was fast like that of acetaldehyde, the reduced decomposition rate would not occur. The decomposition of acetone was found to be a rate-determining step in the overall reaction with UV/photocatalyst. The optimization of preparation conditions with acetone removal instead of isopropyl alcohol would be a challenging, but valuable, approach.

## 4. Conclusions

A photocatalyst was immobilized using the sol-gel based method for decomposing low molecular weight compounds in electronics wastewater for reuse. The amount of precursor and support, water to precursor ratio, pH, aging time, and calcination conditions were optimized by removal of isopropyl alcohol as a representative of target compounds. SEM and XRD analysis results were performed for aiding selection of variables. The preparation conditions of the sol-gel process (amount of support = 15 g, amount of precursor = 12 mL, amount of water = 2.4 mL, and aging time = 60 min) and the subsequent calcination conditions (calcination temperature = 500 °C and calcination time = 2 h) were determined to have a high removal efficiency of isopropyl alcohol. Removal efficiencies of isopropyl alcohol, acetone, ethanol, and acetaldehyde in synthetic wastewater with UV irradiation and photocatalyst reached 97.2%, 71.2%, 99.0%, and 99.0% in 2 h. To better understand the mechanism of photocatalytic decomposition of low molecular weight compounds, the reaction kinetics of the target compound were investigated. Reaction constants were determined by fitting experimental data and the trial and error method with the assumption of first order kinetics with consecutive reaction pathway. The suggested reaction pathway was confirmed and UV with prepared photocatalysts was found effective after comparison of the reaction constants. The decomposition of acetone was found to be a rate-determining step in the overall reaction by UV/photocatalyst. The prepared photocatalysts have great potential in the application of wastewater reuse by decomposition of low molecular weight organic compounds.

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
