# Peer review of "Application of Wastewater Reuse with Photocatalyst Prepared by Sol-Gel Method and Its Kinetics on the Decomposition of Low Molecular Weight Pollutants"

_ijerph, 2020, doi:10.3390/ijerph17124203_

Round 1

Reviewer 1 Report

The authors reported that extensive study on the preparation of photocatalytic media by sol gel method including the amount of precursor and support, water to precursor ratio, pH, aging time, and calcination conditions was performed. The conditions were chosen by measuring removal efficiencies of isopropyl alcohol as a representative target compound, and supportive SEM and XRD analyses. Removal efficiency in synthetic wastewater including isopropyl alcohol, acetone, ethanol, and acetaldehyde was evaluated with time. Removal efficiencies reached 97.2%, 71.2%, 99.0%, and 99.0% in 2 hours. The reaction kinetics of each of the compounds were assumed to be first order with a consecutive reaction pathway. Reaction constants were determined by fitting
experimental data, and the trial and error method. This work is interesting adn can be accepted after proper revision. (1) Why the auhtors choose sol-gel instead of other methods? The authors may compare other methods like Appl Catal. B 2019, 248, 298. Please mention the benefits of this method; (2) advanced oxidation processes like photocatalysis has been widely used in Science of The Total Environment 2020, 704, 135284. What is the nolvety of this work? (3) How about the stability? (4) The proper error bar of the experimnetal data may add. (5) Abstract: please include the future trend. 

Author Response

Reviewer #1:

The authors reported that extensive study on the preparation of photocatalytic media by sol gel method including the amount of precursor and support, water to precursor ratio, pH, aging time, and calcination conditions was performed. The conditions were chosen by measuring removal efficiencies of isopropyl alcohol as a representative target compound, and supportive SEM and XRD analyses. Removal efficiency in synthetic wastewater including isopropyl alcohol, acetone, ethanol, and acetaldehyde was evaluated with time. Removal efficiencies reached 97.2%, 71.2%, 99.0%, and 99.0% in 2 hours. The reaction kinetics of each of the compounds were assumed to be first order with a consecutive reaction pathway.

Reaction constants were determined by fitting experimental data, and the trial and error method. This work is interesting and can be accepted after proper revision.

  1. Why the authors choose sol-gel instead of other methods? The authors may compare other methods like Appl Catal. B 2019, 248, 298. Please mention the benefits of this method;

Compared with CVD and PVD, sol-gel approach is advantageous due to its mild condition, low cost, simple approach as discussed in the literature author recommend. In introduction part, the following sentence is included.

“Photocatalysts have been immobilized on supports, such as glass, sand, Teflon, activated carbon, and ceramics, by the sol-gel method, chemical vapor deposition, and physical vapor deposition [8]. Among these, the sol–gel method is a promising technique for immobilization of photocatalysts due to its mild conditions, simple process, and low cost [9].”

  1. Advanced oxidation processes like photocatalysis has been widely used in Science of The Total Environment 2020, 704, 135284. What is the novelty of this work?

This is the first paper in application of wastewater reuse with TiO2 photocatalyst prepared by sol-gel method.

  1. How about the stability?

Detachment of photocatalyst was observed after calcination of sample with excessive coating layer. In optimized condition, detachment of photocatalyst was not observed after calcination. This photocatalytic media was tested in pilot for 10 days without decrease of removal efficiency (data will be included in another manuscript).

  1. The proper error bar of the experimental data may add.

Error bars were added in Fig. 5.

  1. Abstract: please include the future trend. 

The last part of abstract was revised as follows,

“The immobilized photocatalyst developed in this study would be useful for application of wastewater reuse with high removal efficiencies, mild preparation conditions, and mechanical stability.”

Reviewer 2 Report

I read the manuscript “Preparation of photocatalytic media by sol-gel method and its kinetics on the decomposition of isopropyl alcohol, acetone, ethanol, and acetaldehyde” and I found that its content is original and suitable for publication after major revision.

  • The title and the abstract do not strictly reflect the content. A reader cannot immediately catch the novelty and the main objective of the study. In particular, the definition of “photocatalytic media” is not much clear.
  • More details about the material constituting the ceramic beads should be provided.
  • The n° of variables indicated in table 1 is misleading, since the n° of variables should be the number of changing parameters.
  • Line 109: how much H2O2 is added?
  • In the discussion of the results related to the removal of isopropyl alcohol carried out in various preparation conditions and presented in Fig. 2,the fixed parameters for each changing variable are not reported. It could be useful to report them in a table, for instance. See line 144: which is the fixed amount of precursor?
  • Regarding the SEM characterization, it is not clear if the maximum preparation time is 4 hours or 120 min as reported in table 1. Moreover which is the optimized conditions? Authors should be more explicit.
  • Also in the case of XRD all the fixed parameters must be indicated.
  • Line 145: have the authors considered the eventual contribution of surface area and, in general, surface characteristics? Line 150: can the authors give details about the defects evidenced?
  • In general the discussion from 155 to 166 is quite confusing with some contradictory data. I suggest the author to revise this section, proposing more solid explanations of the phenomena.
  • Line 177: why 12 ml was chosen as the optimized amount of precursor? From fig. 2, 16 ml resulted in the highest abatement of isopropyl alcohol.
  • Why the removal efficiency showed in fig. 2 is so different with respect to the results reported in fig. 5 (in which also a percentage near 100% was reached)? Also in the case of fig. 5, please report the working conditions.
  • Line 217: authors exclude the presence of other kinds of radicals beyond hydroxyl ones?

Author Response

I read the manuscript “Preparation of photocatalytic media by sol-gel method and its kinetics on the decomposition of isopropyl alcohol, acetone, ethanol, and acetaldehyde” and I found that its content is original and suitable for publication after major revision.

  1. The title and the abstract do not strictly reflect the content. A reader cannot immediately catch the novelty and the main objective of the study. In particular, the definition of “photocatalytic media” is not much clear.

We changed title of manuscript to emphasize the novelty of this manuscript.

“Application of wastewater reuse with photocatalyst prepared by sol-gel method and its kinetics on the decomposition of low molecular weight pollutants”

  1. More details about the material constituting the ceramic beads should be provided.

Materials constituting the ceramic beads were included in materials and methods section.

“Ceramic hollow beads (DFL, Minmet Refractories Co, Cenosphere, China) were used as support to bind photocatalysts to in the fluidized bed reactor. These beads were mainly composed alumina (Al2O3) and silica (SiO2) and were 40 – 300 μm in diameter with particle density of 0.9 – 1.0 g/ml.”

  1. The n° of variables indicated in table 1 is misleading, since the n° of variables should be the number of changing parameters.

Full list of preparation conditions (26 trials) were provided in table 1 and no of variables were replaced by No. of trials.

  1. Line 109: how much H2O2 is added?

“Hydrogen peroxide” was erased. UV irradiation was compared with UV irradiation with photocatalyst.  

  1. In the discussion of the results related to the removal of isopropyl alcohol carried out in various preparation conditions and presented in Fig. 2, the fixed parameters for each changing variable are not reported. It could be useful to report them in a table, for instance. See line 144: which is the fixed amount of precursor?

Full list of preparation conditions (26 trials) were provided in table 1 to show fixed variables in each condition.

  1. Regarding the SEM characterization, it is not clear if the maximum preparation time is 4 hours or 120 min as reported in table 1. Moreover which is the optimized conditions? Authors should be more explicit.

Full list of preparation conditions (26 trials) were provided in table 1 to show fixed variables in each condition.

  1. Also in the case of XRD all the fixed parameters must be indicated.

Fixed parameters were included in the caption of Fig. 4.

“Figure 4. Phase structures of photocatalytic media under different: a) calcination temperatures and b) calcination times (prepared in the optimized reaction conditions (amount of water = 2.4 ml, amount of precursor = 12 ml, aging time = 60 min, reaction pH=1, amount of beads = 15 g)”

  1. Line 145: have the authors considered the eventual contribution of surface area and, in general, surface characteristics? Line 150: can the authors give details about the defects evidenced?

We added ‘possibly’ in the text because we did not measure surface area

We added the following sentence.

“A defective layer of photocatalyst was shown in Fig. 3b (aging time = 120 minutes), while defect-less layer in Fig. 3a (aging time = 60 minutes). The detachment of the excessive coating layer was observed after calcination of photocatalyst with more than 60 minutes of aging time.”

  1. In general the discussion from 155 to 166 is quite confusing with some contradictory data. I suggest the author to revise this section, proposing more solid explanations of the phenomena.

We revised this section,

“The amount of precursor, support, and aging time were variables affecting the mechanical stability as well as photochemical ability of photocatalyst.

In addition, the pH and amount of water in the sol-gel reaction plays an important role in controlling the reaction rate of hydrolysis and condensation [14]. Fig. 2d and Fig. 2e indicate that a lower pH and more water addition favor a higher removal. The SEM results in Fig. 3d and Fig. 3f show that a high pH and a more water addition produced agglomeration on the surface of the photocatalyst layer. The water to precursor molar ratio (r = [H2O] / [Ti(OBu)4]) in the sol-gel process strongly affected the morphology of the photocatalyst. It was reported that the Ti(OBu)4 precursor was fully hydrolyzed to Ti(OOH)4 in a high ratio (r = 4), while it was partially hydrolyzed to Ti(OH)r(OBu)4-r in a low r ratio (r < 4). The fully hydrolyzed precursor would lead to a more crosslinked structure [15]. In addition, it has been reported that a high r ratio favored a smaller size of particles, which is in agreement with the results of this study [11]. Amount of water was determined to have r close to 4 (amount of water = 2.4 ml) with less agglomeration.”

  1. Line 177: why 12 ml was chosen as the optimized amount of precursor? From fig. 2, 16 ml resulted in the highest abatement of isopropyl alcohol.

12 ml of precursor was chosen instead of 16ml to avoid agglomeration of photocatalyst.

  1. Why the removal efficiency showed in fig. 2 is so different with respect to the results reported in fig. 5 (in which also a percentage near 100% was reached)? Also in the case of fig. 5, please report the working conditions.

In Fig. 2, initial concentration of IPA was 10 mg/L before optimization of variables

In Fig. 5, initial concentration of IPA was 1 mg/L after optimization of variables.

The caption in fig. 2 and 5 were changed to explain the difference.

“Figure 2. Effects of a) amount of support, b) amount of precursor, c) aging time, d) pH, e) amount of water, f) calcination temperature, and g) calcination time in preparation of immobilized”

” Figure 5. Removal efficiencies of a) isopropyl alcohol, b) acetone, c) ethanol, and d) acetaldehyde by UV and UV + prepared photocatalyst. (Tested under synthetic wastewater with 1 mg/L of isopropyl alcohol, acetone, ethanol, and acetaldehyde, respectively) “

  1. Line 217: authors exclude the presence of other kinds of radicals beyond hydroxyl ones?

Authors included revised the following sentence.

“The reactions between target compounds and hydroxyl radicals involved in the reaction pathway would be described as follows,”

Reviewer 3 Report

This study is the preparation of photocatalyst media, in particular, in terms of lots of possible experimental variation to improve the removal efficiency of the system. All the experimental data is well described and organized through the manuscript. However, a few point of the data and its interpretation should be revised before considering the publication. Detail comments are listed bellows.

  1. It is recommended to insert any schemes for the preapred photocatalyst media, for example, something like ‘Figure 5 in M.-H Baek et al. Applied Catalysis A: general 450 (2013) 222-229’. This would be helpful for readers to see what exactly author newly developed for the required system. All the SEM images in the manuscript shows very little portion of whole system.
  2. Page 8 line 160, author explained the effect of water addition on the morphology of the photocatalyst with respect to ratio, r. However, in the figure 2e only showed the removal efficiency with various of the amount of water, not with the ratio. If possible, it would be better to add the ratio of each water amount as additional x-axis.
  3. In the SEM image of Figure 3f, the surface looked more agglomerated compared to that of Figure 3e. However, the removal efficiency showed rather higher in such agglomerated samples (Figure 2e). Author should show the reason because the SEM image of Figure 3e seems to have more specific surface area for catalytic reaction.
  4. In the Figure 3h, it would be best to mark where it is additional photocatalyst layer or not by arrow or guideline for easy understanding .

  5. In the Figure 5, author should conduct the same removal experimental in the media without UV, which is important to determine those efficiency by adsorption on the media surface.

Author Response

This study is the preparation of photocatalyst media, in particular, in terms of lots of possible experimental variation to improve the removal efficiency of the system. All the experimental data is well described and organized through the manuscript. However, a few points of the data and its interpretation should be revised before considering the publication. Detail comments are listed bellows.

  1. It is recommended to insert any schemes for the preapred photocatalyst media, for example, something like ‘Figure 5 in M.-H Baek et al. Applied Catalysis A: general 450 (2013) 222-229’. This would be helpful for readers to see what exactly author newly developed for the required system. All the SEM images in the manuscript shows very little portion of whole system.

As reviewer recommends, scheme for immobilized photocatalyst was included in scheme1.

  1. Page 8 line 160, author explained the effect of water addition on the morphology of the photocatalyst with respect to ratio, r. However, in the figure 2e only showed the removal efficiency with various of the amount of water, not with the ratio. If possible, it would be better to add the ratio of each water amount as additional x-axis.

Fig. 2e was revised to show ratio of [H2O]/[Ti(OBu)4] as x-axis.

  1. In the SEM image of Figure 3f, the surface looked more agglomerated compared to that of Figure 3e. However, the removal efficiency showed rather higher in such agglomerated samples (Figure 2e). Author should show the reason because the SEM image of Figure 3e seems to have more specific surface area for catalytic reaction.

Arrangement in fig 3 was corrected. Location of 3c & 3d were switched with 3e & 3f

  1. In the Figure 3h, it would be best to mark where it is additional photocatalyst layer or not by arrow or guideline for easy understanding.

We added arrow for a newly-formed photocatalyst layer on support in figure 3h and caption in Figure 3.

“Figure 3. Surface morphologies of photocatalytic media prepared under different conditions (a: aging time = 60 minutes, b: aging time = 120 minutes, c: pH = 1, d: pH = 7, e: amount of water =, 0.6 mL, f: amount of water = 3.6 mL) and those prepared in optimized conditions in different magnifications (g and h, arrow for deposition of photocatalyst layer in (h)).”

  1. In the Figure 5, author should conduct the same removal experimental in the media without UV, which is important to determine those efficiency by adsorption on the media surface.

Unfortunately, dark adsorption of prepared photocatalyst was not tested in this study. Instead, we focused on baseline correction included in the last section of materials and methods, as follows,

“Standard solutions for target compounds were prepared by series dilution of stock solution prepared by ACS grade chemicals (Sigma Aldrich). Deviation from calibration curve was checked in every 3-5 samples and kept less than 10%. The concentrations of target compounds in the blank eluent from hollow beads, dissolved in water from air, and in distilled water were measured to correct the baseline as described in the literature [13].”

Reviewer 4 Report

Well, I have couple of questions. 

authors used gcms to obtain data. What standard did they use to obtain amount left for any organic molecules.

second question is what standard experiment they use?

Author Response

Well, I have couple of questions. 

  1. Authors used gcms to obtain data. What standard did they use to obtain amount left for any organic molecules.

The following sentences describing our in-house method were included,

“Standard solutions for target compounds were prepared by series dilution of stock solution prepared by ACS grade chemicals (Sigma Aldrich). Deviation from calibration curve was checked in every 3-5 samples and kept less than 10%. The concentrations of target compounds in the blank eluent from hollow beads, dissolved in water from air, and in distilled water were measured to correct the baseline as described in the literature [13].”

  1. Second question is what standard experiment they use?

Because there is no standard method, we used our in-house analysis method for extremely low concentration which includes baseline check with air, distilled water, and support. This method was provided in a publication from one of authors’ laboratory. (Science of Advanced Materials, Vol. 11, pp. 1–9, 2019).

The following sentences was included to explain our in-house method,

“Standard solutions for target compounds were prepared by series dilution of stock solution prepared by ACS grade chemicals (Sigma Aldrich). Deviation from calibration curve was checked in every 3-5 samples and kept less than 10%. The concentrations of target compounds in the blank eluent from hollow beads, dissolved in water from air, and in distilled water were measured to correct the baseline as described in the literature [13].”

Round 2

Reviewer 2 Report

The revised version is substantially improved and the authors provided the suggested modifications. After a further careful revision of the english form, the paper is suitable for publication.

Reviewer 3 Report

As all the comments and suggestion was now well reflected to current revised manuscript by authors, I'd agree this manuscript to be good for publication in IJERPH.